# Epstein-Barr Virus DNAemia and post-transplant lymphoproliferative disorder in pediatric solid organ transplant recipients

Yeh-Chung Chang[1]*, Rebecca R. Young[1], Alisha M. Mavis[2], Eileen T. Chambers[3], Sonya Kirmani[4], Matthew S. Kelly[1], Ibukunoluwa C. Kalu[1], Michael J. Smith[1], Debra J. Lugo[1]

1 Department of Pediatrics, Division of Infectious Diseases, Duke University School of Medicine, Durham, North Carolina, United States of America, 2 Department of Pediatrics, Division of Gastroenterology, Duke University School of Medicine, Durham, North Carolina, United States of America, 3 Department of Pediatrics, Division of Nephrology, Duke University School of Medicine, Durham, North Carolina, United States of America, 4 Department of Pediatrics, Division of Cardiology, University of Wisconsin-Madison, Madison, Wisconsin, United States of America

* dan.chang@duke.edu

**Data Availability Statement:** All relevant data are within the manuscript and its Supporting Information files.

## Abstract

### Background

Pediatric solid organ transplant (SOT) recipients commonly have Epstein-Barr virus (EBV) DNAemia and are at risk of developing post-transplant lymphoproliferative disorder (PTLD). EBV DNAemia has not been analyzed on a continuous scale in this population.

### Methods

All children $\leq$ 18 years of age who underwent SOT at a single center between January 1, 2007 and July 31, 2018 were included in this retrospective study. Transplant episodes in which PTLD occurred were compared to transplant episodes without PTLD. Multivariable logistic regression was used to identify factors associated with the development of EBV DNAemia and maximum height of EBV DNAemia. A Cox proportional hazards model was used to calculate hazard ratios for time to PTLD.

### Results

Of 275 total transplant recipients and 294 transplant episodes, there were 14 episodes of PTLD. Intestinal and multivisceral transplant were strongly associated with PTLD (p = 0.002). Risk factors for the development of EBV DNAemia include donor and recipient positive EBV serologies (p = 0.001) and older age (p = 0.001). Maximum level of EBV DNAemia was significantly associated with development of PTLD (p<0.0001). Every one log ($\log_{10}$) increase in the maximum level of EBV DNAemia was associated with a more than doubling of the hazard on developing PTLD (HR: 2.18, 95% CI 1.19–3.99).

### Conclusions

Transplant type was strongly associated with development of PTLD in pediatric SOT recipients. EBV serologies and age were associated with the development of EBV DNAemia and

**Funding:** The author(s) received no specific funding for this work.

**Competing interests:** The authors have declared that no competing interests exist.

height of DNAemia. High levels of EBV DNAemia were strongly associated with an increased hazard for PTLD.

## Introduction

Post-transplant lymphoproliferative disorder (PTLD) is a significant cause of morbidity and mortality in pediatric solid organ transplantation (SOT) recipients. Epstein-Barr Virus (EBV) DNAemia plays a key role in the pathophysiology of EBV-related PTLD but characteristics of pediatric SOT recipients who develop EBV DNAemia and patterns of EBV DNAemia have not been well defined [1]. Many centers monitor EBV DNAemia with established protocols for interventional measures such as decreasing immunosuppression [2]. These protocols may differ between centers and between different types of organ transplant recipients [3]. More data is needed to guide clinical decisions in specific scenarios.

There have been many studies on risk factors for PTLD in SOT recipients [1–4]. Overall, risk factors for development of PTLD within the first 12 months of transplant (early PTLD) include: type of organ transplant, T-cell depleting therapies such as anti-thymocyte globulin (ATG), and young age. Early PTLD is also strongly influenced by donor and recipient serologies as donor-derived EBV infection is a strong risk factor for PTLD. Risk factors for development of PTLD after 12 months post-transplant (late PTLD) include length and duration of immunosuppression, and type of organ transplant. Intestinal transplant recipients have been found to have the highest risk for PTLD, followed by lung, then heart, then liver, and lastly, kidney transplant recipients.

There is a higher incidence of PTLD in pediatric SOT recipients when compared to adult SOT recipients [1]. One reason is that children are at higher risk for primary infection, and primary EBV infection has a higher risk of causing PTLD [4]. One pediatric SOT study found that EBV viral load was significantly higher in cases of PTLD, when matched to controls [5]. Another study reviewing only pediatric patients found the following risk factors to be significant in a univariate model: age at transplant, use of basiliximab, steroid intensification, ATG, median peak EBV value level, and chronically elevated EBV DNAemia [6]. However, in the multivariable model, only age, use of steroids, and peak EBV value were significant.

Many previous studies in pediatric solid organ transplants have categorized EBV DNAemia into three categories: 1) chronic high load (CHL) carriers, 2) chronic low viral load carriers, and 3) those with sustained undetectable EBV levels, with chronic defined as a time period > 6 months [7–10]. In pediatric heart transplant recipients, CHL carriers are predisposed to PTLD, and pediatric intestinal transplant recipients are at intermediate risk [7, 8]. In the pediatric liver and kidney populations, CHL carriers are not at higher risk of PTLD [9, 10]. Despite low sample sizes within studies, this difference suggests that transplant type and immunosuppression play a large role in the development of PTLD.

There have been many studies on trying to categorize cutoffs for EBV DNAemia and risk for PTLD, as well as establishing a predictive model for PTLD [6, 11, 12]. While many of these studies have tried to categorize a certain EBV quantitative cutoff level for when patients are at risk for PTLD, there are confounding risk factors such as patient age, type of transplant and type of immunosuppression and not all models have been well validated. In addition, there is controversy between the use of whole blood vs plasma samples, and there is interlaboratory variability for quantification of EBV levels from the same samples [13].

The epidemiology of PTLD has been changing in recent years [14, 15]. In addition, EBV monitoring has provided a wealth of data that has not been studied in a detailed manner.

While EBV DNAemia has been categorized in previous studies, it has not yet been studied on a continuous scale and available pediatric literature has not linked maximum height of EBV DNAemia to PTLD on a granular level. Therefore, we undertook a comprehensive retrospective descriptive study at a large pediatric SOT center to specifically evaluate risk factors associated with EBV DNAemia. We also wanted to investigate the predictive value of continuous EBV monitoring for PTLD in order to demonstrate its ongoing utility.

## Material and methods

We performed a retrospective study of patients ≤ 18 years of age who received a solid organ transplant at Duke University Medical Center between January 1, 2007 and July 31, 2018. Events were recorded by transplant episode. The population included heart, kidney, liver, lung, intestinal, and multivisceral transplant recipients. Characterization of the pediatric SOT population was performed using the United Network for Organ Sharing (UNOS) database, and data was also extracted from the center's electronic health records using the program Duke Enterprise Data Unified Content Explorer (DEDUCE) [16]. The diagnosis of PTLD and lymphoma were identified by diagnosis code. The diagnoses were next verified through chart review of pathology results and progress notes in the electronic health record. The study was approved and granted a waiver of informed consent by the Duke University institutional review board (IRB). Patient health information and identifiers were available within a secure workspace, but only de-identified, anonymized data can be exported out of this workspace.

At our center, each pediatric solid organ transplant program has developed their own methods of screening for EBV DNAemia. However, in general, quantitative whole blood EBV polymerase chain reaction (PCR) studies are done monthly for the first year post-transplant, then spaced out to every 3 months for the second and third year post-transplant, then either every 6 months or yearly afterwards if there are no issues. If EBV DNAemia was detected, quantitative PCRs were obtained more frequently. All EBV PCRs included in this study were done at the central laboratory of our center, which used the same machine over the study period, the ABI PRISM 7500/7500DX Sequence Detection System (Applied Biosystems) and Qiagen EBV PCR (ASR) reagents.

EBV DNAemia values were extracted from the electronic health record, including accompanying chart review for missing values. For most of the study period, the quantitative unit of EBV DNAemia was reported as copies/μL. Starting in mid-2017, this was changed to international units/mL (IU/mL), and a conversion factor of 1 copy/μL = 113.6 IU/mL was used to normalize these values in concordance with the previous quantification. The laboratory checks this conversion every 6 months and the conversion factor has not changed over the duration of the study.

Patients who developed EBV DNAemia after transplant were compared to patients who did not develop DNAemia. EBV DNAemia was defined as any detection of EBV virus by quantitative polymerase reaction (PCR). EBV donor and recipient serologies were based on what was entered into the patient's chart at time of transplant. Demographic and clinical characteristics were compared between the two groups, using Chi square and Fisher's exact tests for categorical factors and Mann-Whitney U tests for continuous factors. A logistic regression was done to calculate odds of EBV DNAemia using the variables of age, race, type of transplant, type of induction immunosuppression, and donor and recipient EBV serologies. These variables were either significant in the unadjusted model or risk factors that were denoted to be clinically significant a priori based on current knowledge.

The diagnosis of PTLD was confirmed by chart review. All cases of PTLD had accompanying histologic diagnoses which included EBER staining and description of atypical cells seen

on pathology. Sources of biopsy tissue included lymph nodes, tonsils and adenoids, and the transplanted organ.

For the analysis of the height of EBV DNAemia, the maximum single EBV value was identified. That value was then $\log_{10}$ transformed and an analysis of covariance (ANCOVA) was used to evaluate associations between the maximum EBV value and demographic and clinical factors. After categorization and taking the average of the $\log_{10}$ transformed values, the data was back transformed, establishing the geometric mean. Logarithmic comparison of quantitative EBV values has been well described in the literature [17, 18].

A Cox proportional hazards model was used to link clinical risk factors to time to PTLD. The event of PTLD, death, and the event of a subsequent transplant were used as censoring criteria. The last observed value was carried forward. A robust variance (sandwich) estimator was used to control for the fact that some children were included more than once, due to multiple transplants. Adjusted models incorporated variables that were significant in the unadjusted model as well as risk factors that were denoted to be clinically significant a priori.

## Results

We reviewed 275 pediatric patients who underwent 294 transplant episodes (Table 1). The median (interquartile range) age of patients was 4 (IQR 0, 14) years. The most common transplant type was liver (46%), followed by heart (25%), kidney (13%), multivisceral (7%), lung (7%), and intestinal (3%). For EBV serologies, donor positive/recipient negative (D+/R-) and donor positive/recipient positive (D+/R+) were the most common (33% and 31% respectively), followed by donor negative/recipient negative (D-/R-) (14%), and then donor negative/recipient positive (D-/R+) (11%), with the remaining unknown (11%). A total of 55 children (18.7%) received T-cell depleting therapy including anti-thymocyte globulin (ATG) or alemtuzumab (Campath) at induction.

All children who developed PTLD were EBV positive by quantitative PCR, and thus, only EBV positive children were included in following analyses (Table 2). Fourteen children developed PTLD after transplant, for an overall incidence rate of 4.8%. Transplant type was significantly associated with PTLD (p = 0.001). Intestinal and multivisceral transplant recipients accounted for 21% and 14% of PTLD cases respectively, while only accounting for 2% and 3% of the total population that was EBV positive. There was also an association between PTLD and race (p = 0.04) as there were a total of six Asian children in the study who were EBV positive, and two developed PTLD. Asian children comprised 14% of the PTLD cases, but only 2.2% of the study population. Induction immunosuppression and age were not associated with PTLD in our study (p = 0.18 and p = 0.17, respectively). For the nine patients who developed PTLD within the first year, EBV values were plotted over time (Fig 1). For most patients, the general pattern was a steady increase in EBV values, followed by diagnosis of PTLD, and then a rapid decline.

Risk factors for EBV DNAemia were examined (Table 3). Significant risk factors for EBV DNAemia included: EBV serology at the time of transplant, age, and type of transplant in the unadjusted model, and only EBV serology and age in the adjusted model (Table 4). Heart transplant recipients were at lower odds for EBV DNAemia, with OR 0.44 (95% CI 0.22–0.89). Lung transplant recipients were similarly at lower odds, but sample size was limited. Donor positive/recipient negative (D+/R-) and donor positive/recipient positive (D+/R+) transplant recipients were at increased odds of developing EBV DNAemia compared to donor negative/recipient negative (D-/R-) transplant recipients, with OR 3.90 (95% CI 1.55–9.80), and OR 4.83 (95% CI 2.02–11.55), respectively. Older children were also associated with an increased risk, with every one year increase in age associated with an OR of 1.10 (1.04–1.16).

**Table 1. Characteristics of all children in the study, regardless of PTLD or EBV status.**

| | | n | % |
|---|---|---|---|
| Total number of transplants | | 294 | |
| Total number of children | | 275 | |
| Race/Ethnicity | | | |
| | American Indian or Alaska Native | 3 | 1.1% |
| | Asian | 6 | 2.2% |
| | Black | 84 | 30.6% |
| | Hispanic | 35 | 12.7% |
| | Multiracial | 6 | 2.2% |
| | White | 141 | 51.3% |
| Gender | | | |
| | Male | 148 | 53.8% |
| | Female | 127 | 46.2% |
| Type of Organ Transplant | | | |
| | Heart | 73 | 24.8% |
| | Intestine | 10 | 3.4% |
| | Kidney | 38 | 12.9% |
| | Liver | 134 | 45.6% |
| | Lung | 19 | 6.5% |
| | Multivisceral | 20 | 6.8% |
| EBV Serology at Transplant | | | |
| | D+/R+ | 98 | 33.3% |
| | D+/R- | 92 | 31.3% |
| | D-/R+ | 32 | 10.9% |
| | D-/R- | 40 | 13.6% |
| | Unknown | 32 | 10.9% |
| Transplant Number | | | |
| | First | 274 | 93.2% |
| | Second | 16 | 5.4% |
| | Third | 4 | 1.4% |
| EBV Status Post Transplant (Positive EBV PCR) | | | |
| | EBV Positive | 160 | 54.4% |
| | EBV Negative | 111 | 37.8% |
| | No testing | 23 | 7.8% |
| Induction Immunosuppression | | | |
| | ATG or T-cell depleting therapy | 55 | 18.7% |
| | Not ATG or T-cell depleting therapy | 231 | 78.6% |
| | Missing | 8 | 2.7% |
| | median | | (p25, p75) |
| Year of Transplant | | 2013 | (2011, 2016) |
| Age at Transplant (years) | | 4 | (0, 14) |

An analysis of the geometric mean of the highest value of EBV DNAemia (Table 5) showed that in the unadjusted model, liver transplant recipients had a higher geometric mean of the maximum EBV value 79.8 units/uL (95% CI 47–135.5) compared to heart transplant recipients at 14.6 units/uL (95% CI 6.6–32.1), and kidney transplant recipients 8.6 units/uL (95% CI 3.3–22.6), p = 0.007 and p<0.001 respectively. In the adjusted model, liver transplant recipients had a higher geometric mean of the maximum EBV value at 53 units/µL (95% CI 26.3, 106.9)

**Table 2. The characteristics of children who had PTLD at any time post-transplant are compared to children who had no PTLD.** This is limited to the children who had any EBV DNAemia after transplant. The incidence of PTLD is shown for each demographic group.

| | | PTLD | | No PTLD | | |
|---|---|---|---|---|---|---|
| | | n | % | n | % | p-value |
| Total | | 14 | | 146 | | |
| Type of Organ Transplant | | | | | | |
| | Heart | 1 | 7.1% | 36 | 24.7% | 0.001*+ |
| | Intestine | 3 | 21.4% | 3 | 2.1% | |
| | Kidney | 3 | 21.4% | 22 | 15.1% | |
| | Liver | 4 | 28.6% | 77 | 52.7% | |
| | Lung | 1 | 7.1% | 3 | 2.1% | |
| | Multivisceral | 2 | 14.3% | 5 | 3.4% | |
| EBV Serology at Transplant | | | | | | |
| | D+/R+ | 4 | 28.6% | 53 | 36.3% | 0.47 |
| | D+/R- | 4 | 28.6% | 56 | 38.4% | |
| | D-/R+ | 1 | 7.1% | 14 | 9.6% | |
| | D-/R- | 2 | 14.3% | 10 | 6.8% | |
| | Unknown | 3 | 21.4% | 13 | 8.9% | |
| Race/Ethnicity | | | | | | |
| | American Indian/Alaskan | 0 | 0% | 5 | 3.4% | 0.04*+ |
| | Asian | 2 | 14.3% | 2 | 1.4% | |
| | Black | 2 | 14.3% | 50 | 34.2% | |
| | Hispanic | 3 | 21.4% | 19 | 13.0% | |
| | Multiracial | 0 | 0% | 4 | 2.7% | |
| | White | 7 | 50.0% | 66 | 45.2% | |
| Gender | | | | | | |
| | Female | 3 | 21.4% | 67 | 45.9% | 0.078 |
| | Male | 11 | 78.6% | 79 | 54.1% | |
| Transplant Number | | | | | | |
| | 1 | 14 | 100% | 133 | 91.1% | 0.51+ |
| | 2 | 0 | 0% | 10 | 6.9% | |
| | 3 | 0 | 0% | 3 | 2.1% | |
| Induction Immunosuppression | | | | | | |
| | ATG or T-cell depleting therapies | 4 | 28.6% | 26 | 15.4% | 0.18 |
| | Not ATG or T-cell depleting therapies | 9 | 64.3% | 136 | 80.5% | |
| | Unknown/Missing | 1 | 7.1% | 7 | 4.1% | |
| Age (years) at transplant | Median (IQR) | 2 (0, 9) | | 4 (0, 14) | | 0.17 |

*p<0.05

+ By Fisher's-Exact Test

compared to intestinal transplant recipients at 6.4 units/µL (95% CI 1.2, 31.0), p = 0.005. Lastly, children who developed PTLD had higher geometric mean values of EBV, 211.6 (56.7, 790.1) copies/µL, compared to children without PTLD, who had a geometric mean EBV value of 27.3 (18.0, 41.2) copies/µL (p<0.0001). Type of transplant was not significant in the adjusted model (p = 0.14).

A final analysis was performed evaluating the association between risk factors and time to PTLD, and the adjusted analysis is included (Table 6). Every $\log_{10}$ increase in maximum EBV value was associated with a hazards ratio (HR) of 2.18 for development of PTLD (95% CI

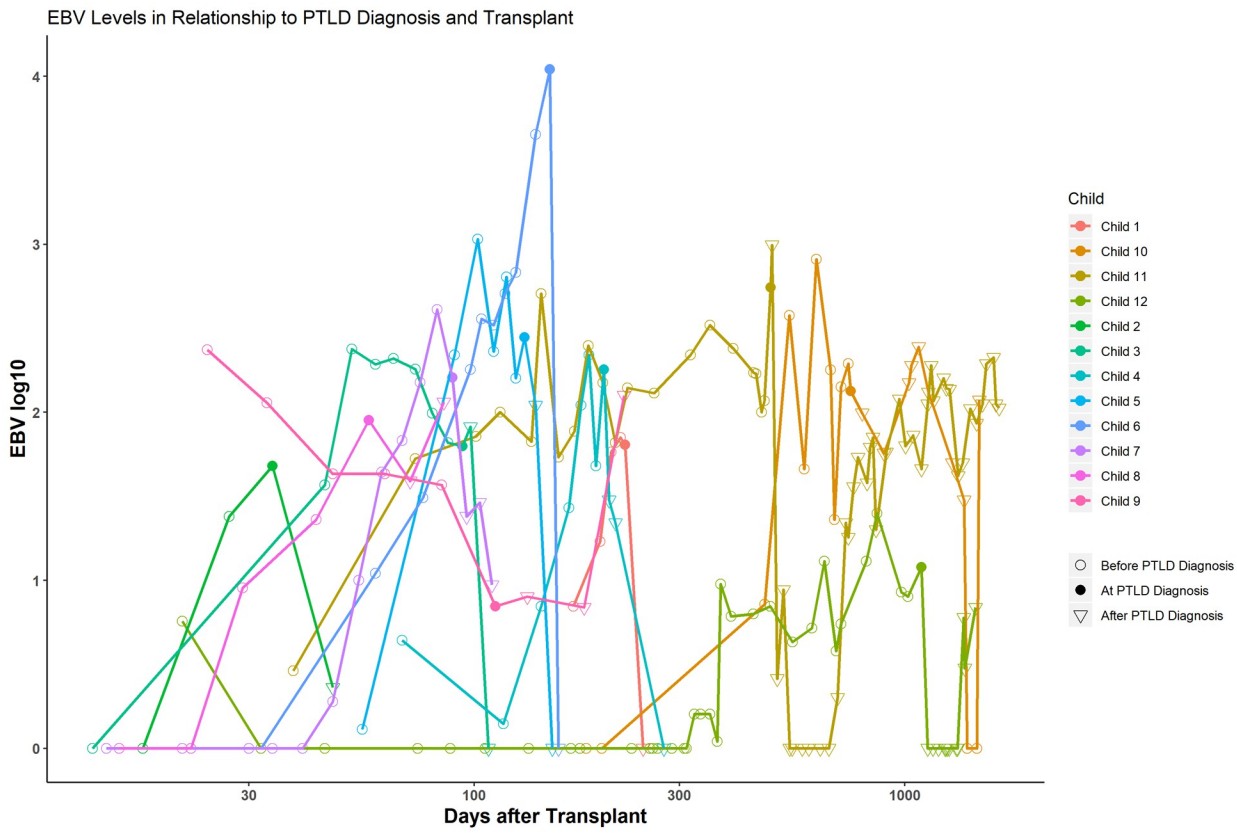

**Fig 1.**

1.19–3.99, p = 0.02). Transplant year was also significant in this analysis. For every one year increase in transplant year, there was a decrease in the hazard ratio, denoted at 0.71 (95% CI 0.53–0.95, p = 0.02). Kidney transplant recipients also had higher hazards ratios for PTLD compared to liver transplant recipients, HR 6.15 (95% CI 1.36–27.73, p = 0.02). There was also a trend toward association between type of induction immunosuppression, with ATG and high dose steroids, with a hazards ratio of 10.1 (95% CI 0.994–103.8, p = 0.0506), which was just out of the range of significance.

Additional data on testing broken down by type of organ transplant and by year is available (Tables 7 and 8). The trends in these tables show that multivisceral and intestinal transplant recipients were tested the most frequently. The number of transplant recipients who were EBV positive by PCR post-transplant did not seem to change over time. In addition, while the number of tests for liver transplant recipients during the first year decreased over the study time period, the number of tests done during the first year for non-liver transplant recipients (heart, kidney, lung, intestinal, and multivisceral transplants) combined did increase overall which suggests better adherence to EBV PCR monitoring over time.

## Discussion

This is the first study to fully describe characteristics of post-transplant EBV DNAemia in the pediatric solid organ transplant population. In a raw analysis of 275 SOT pediatric recipients, type of transplant and race were significantly associated with higher risk for PTLD. Intestinal and multivisceral transplant recipients had the highest risk of developing PTLD, which is

**Table 3. The characteristics of children who had any EBV DNAemia after transplant are compared to those who did not have any EBV DNAemia.**

| | | EBV | | No EBV | | p-value |
|---|---|---|---|---|---|---|
| | | N | % | n | % | |
| Total | | 160 | 59% | 111 | 41% | |
| Type of Organ Transplant | | | | | | |
| | Heart | 37 | 23.1% | 31 | 27.9% | 0.14+ |
| | Intestine | 6 | 3.8% | 3 | 2.7% | |
| | Kidney | 25 | 15.6% | 12 | 10.8% | |
| | Liver | 81 | 50.6% | 47 | 42.3% | |
| | Lung | 4 | 2.5% | 7 | 6.3% | |
| | Multivisceral | 7 | 4.4% | 11 | 9.9% | |
| EBV Serology at Transplant | | | | | | |
| | D+/R+ | 57 | 35.6% | 29 | 26.1% | 0.001* |
| | D+/R- | 60 | 37.5% | 28 | 25.2% | |
| | D-/R+ | 15 | 9.4% | 15 | 13.5% | |
| | D-/R- | 12 | 7.5% | 26 | 23.4% | |
| | Unknown | 16 | 10.0% | 13 | 11.7% | |
| Race | | | | | | |
| | American Indian/Alaskan | 5 | 3.1% | 0 | 0.0% | 0.11 |
| | Asian | 4 | 2.5% | 0 | 0.0% | |
| | Black | 52 | 32.5% | 33 | 29.7% | |
| | Hispanic | 22 | 13.8% | 14 | 12.6% | |
| | Multiracial | 4 | 2.5% | 1 | 0.9% | |
| | White | 73 | 45.6% | 63 | 56.8% | |
| Sex | | | | | | |
| | Female | 70 | 43.8% | 50 | 45.0% | 0.83 |
| | Male | 90 | 56.3% | 61 | 55.0% | |
| Transplant Number | | | | | | |
| | 1 | 147 | 91.9% | 106 | 95.5% | 0.28 |
| | 2 | 10 | 6.3% | 5 | 4.5% | |
| | 3 | 3 | 1.9% | 0 | 0.0% | |
| Induction Immunosuppression | | | | | | |
| | ATG or High Dose Steroids | 27 | 17.0% | 25 | 22.5% | 0.26 |
| | Not ATG or High Dose Steroids | 131 | 83.0% | 86 | 77.5% | |
| | | Median | (p25, p75) | median | (p25, p75) | |
| Year of Transplant | | 2014 | (2011, 2016) | 2012 | (2010, 2015) | 0.58 |
| Age at Transplant (years) | | 7 | (1, 14) | 1 | (0, 10) | 0.001* |

*p<0.05

+By Fisher's-Exact Test

supported by the literature. We interpret the findings of race with caution, as the overall sample size is small, and only two out of four Asians with EBV DNAemia developed PTLD. We were unable to find any other risk factors that were significantly associated with PTLD in our population, including type of induction immunosuppression, EBV donor and recipient serologies, or age. This may have been due to distinctions within our population or to limited sample size.

When we examine risk factors for EBV DNAemia, and for magnitude of EBV DNAemia, some common themes emerge. For EBV DNAemia, EBV serologies of the donor and the

**Table 4. The association between demographic characteristics and EBV DNAemia.** Only significant risk factors are included in this table.

| | | EBV (n) | No EBV (n) | p-value | OR of EBV DNAemia (95% CI) | p-value |
|---|---|---|---|---|---|---|
| Total | | 160 | 111 | | | |
| Type of Organ Transplant | | | | | | |
| | Heart | 37 | 31 | 0.14 | 0.44 (0.22, 0.89) | 0.02 |
| | Intestine | 6 | 3 | | 2.07 (0.44, 9.76) | 0.36 |
| | Kidney | 25 | 12 | | 0.57 (0.22, 1.43) | 0.23 |
| | Liver | 81 | 47 | | Reference | |
| | Lung | 4 | 7 | | 0.08 (0.02, 0.36) | 0.001 |
| | Multivisceral | 7 | 11 | | 0.47 (0.16, 1.36) | 0.16 |
| EBV Serology at Transplant | | | | | | |
| | D+/R+ | 57 | 29 | 0.001 | 3.90 (1.55, 9.80) | 0.004 |
| | D+/R- | 60 | 28 | | 4.83 (2.02, 11.55) | 0.0005 |
| | D-/R+ | 15 | 15 | | 2.48 (0.85, 7.19) | 0.096 |
| | D-/R- | 12 | 26 | | Reference | |
| | Unknown | 16 | 13 | | 2.50 (0.87, 7.17) | 0.089 |
| | | Median (IQR) | Median (IQR) | | | |
| Age at Transplant | | 5.6 (0.8, 15.1) | 4.1 (0.8, 13.8) | | 1.10 (1.04, 1.16) (Per Every One Year Increase) | 0.0005 |

recipient are an obvious risk factor. Age is also a practical risk factor as the risk of being exposed to EBV increases with age. Type of transplant is not associated with risk of EBV DNAemia in the adjusted model, but within the category, we see that EBV DNAemia tended to be more common in liver transplant recipients.

**Table 5. The geometric mean maximum EBV value (Mean of Max EBV) in copies/uL is shown for each demographic group in an unadjusted model, and an adjusted model.**

| | | Unadjusted Model | | | Adjusted Model | | |
|---|---|---|---|---|---|---|---|
| | | Mean of Max EBV | 95% CI | p-value | Mean of Max EBV | 95% CI | p-value |
| Type of Organ Transplant | | | | | | | |
| | Heart | 14.6 | (6.6, 32.1) | 0.007 | 39.8 | (17.7, 89.5) | 0.99 |
| | Intestine | 8.2 | (1.2, 56.7) | 0.224 | 6.4 | (1.2, 31.0) | 0.005 |
| | Kidney | 8.6 | (3.3, 22.6) | <0.001 | 34.5 | (13.3, 89.3) | 0.17 |
| | Liver | 79.8 | (47, 135.5) | Ref. | 53 | (26.3, 106.9) | Ref. |
| | Lung | 7.4 | (0.5, 114.7) | 0.095 | 60 | (5.0, 722.3) | 0.5 |
| | Multivisceral | 50 | (8.3, 300) | 0.622 | 20.6 | (4.4, 97.0) | 0.58 |
| EBV Serology at Transplant | | | | | | | |
| | D+/R+ | 11.2 | (5.9, 21.2) | 0.88 | 14.9 | (6.5, 34.5) | 0.605 |
| | D+/R- | 100.3 | (54.1, 185.9) | 0.056 | 62.8 | (29.0, 136.0) | 0.07 |
| | D-/R+ | 42.9 | (12.1, 152.5) | 0.52 | 45.6 | (13.4, 155.0) | 0.26 |
| | D-/R- | 23.3 | (5.9, 91.4) | Ref. | 23.2 | (6.3, 84.9) | Ref. |
| | Unknown | 21.7 | (6.6, 71) | 0.94 | 20 | (6.6, 60.9) | 0.82 |
| Age at Transplant (For every 1 year increase) | | 0.8 | (0.76, 0.84) | <0.001 | 0.81 | (0.76, 0.86) | <0.001 |
| Induction Immunosuppression | | | | | | | |
| | ATG or High Dose Steroids | 22.4 | (8.5, 59.3) | Ref. | 47 | (17.5, 126.1) | Ref |
| | Not ATG or High Dose Steroids | 34.1 | (21.8, 53.3) | 0.44 | 17.7 | (9.32, 33.5) | 0.1 |
| PTLD | | | | | | | |
| | PTLD + | 211.6 | (56.7, 790.1) | Ref. | 177.6 | (58.6, 538.0) | Ref. |
| | PTLD - | 27.3 | (18.0, 41.2) | <0.001 | 15.2 | (7.4, 31.2) | <0.001 |

**Table 6. Adjusted hazards ratios (HR) for significant covariates for the outcome time to PTLD.**

| Covariate | | Reference | HR | 95% CI | p-value |
|---|---|---|---|---|---|
| EBV DNAemia level | | For every $\log_{10}$ increase in max level of EBV DNAemia | 2.18 | (1.19, 3.99) | 0.02 |
| EBV Serology | | | | | |
| | D+/R+ | D-/R- | 0.37 | (0.05, 2.92) | 0.34 |
| | D+/R- | D-/R- | 0.21 | (0.02, 3.00) | 0.21 |
| | D-/R+ | D-/R- | N/A | | |
| Type of Organ Transplant | | | | | |
| | Heart Transplant | Liver Transplant | N/A | | |
| | Intestine Transplant | Liver Transplant | 1.34 | (0.29, 6.25) | 0.72 |
| | Kidney Transplant | Liver Transplant | 6.15 | (1.36, 27.73) | 0.02 |
| | Lung Transplant | Liver Transplant | 43.58 | (0.97, 1957) | 0.051 |
| | Multivisceral Transplant | Liver Transplant | 0.2 | (0.01, 2.93) | 0.24 |
| Transplant Year | | For every year increase | 0.71 | (0.53, 0.95) | 0.02 |
| Age at transplant (years) | | | 0.98 | (0.85, 1.13) | 0.78 |
| Induction Immunosuppresion | ATG or high dose steroids | No T-cell depleting agents | 10.1 | (0.994, 103.8) | 0.0506 |

Age was the most resilient predictor for the maximum height of EBV DNAemia, as it changed the least from the unadjusted to the adjusted model. Intestinal transplants did not have a significantly different height of EBV DNAemia, yet they still had a higher risk association with PTLD, which suggests that other factors such as length and duration of immunosuppression should be considered. Ultimately, those who developed PTLD had significantly higher maximum levels of EBV DNAemia, which has been seen in previous studies.[5] However, stratifying by type of organ transplant does matter; liver transplant recipients were more likely to have a higher maximum level of DNAemia compared to heart and intestinal transplant recipients but were not more likely to develop PTLD. This may be due to immunosuppression or organ-specific risk factors for PTLD. Interestingly, the association of EBV serostatus D+/R- is often associated with the highest risk for PTLD, and our findings in Tables 2 and 6 do not reinforce this. This may have been limited by sample size, or other unmeasurable factors in our population.

Our final analysis is significant in that higher levels of EBV DNAemia are associated with decreased time to PTLD, even when adjusted for other risk factors. It is also important to note that an earlier date of transplant was associated with a higher hazards ratio for PTLD. This may have been due to better EBV monitoring over time and the development of protocols which have standardized practice at our center. The significance of a higher hazards ratio for kidney transplants is not clear at this time. We did have much more liver transplant recipients

**Table 7.**

| Total number of EBV tests 4614 | | | Median number of tests per patient 12 | | | |
|---|---|---|---|---|---|---|
| Type of organ transplant | Total number of subjects | Missing | Deceased < 1 yr post-transplant | Median number of tests for all patients | Median number of tests for subjects EBV+ by PCR | Median number of tests for subjects EBV- by PCR |
| Heart | 73 | 5 | 5 | 11 | 12.5 | 7 |
| Intestine | 10 | 1 | 0 | 43 | 50 | 27 |
| Kidney | 38 | 1 | 0 | 10 | 10.5 | 10 |
| Liver | 134 | 5 | 1 | 14 | 19 | 8 |
| Lung | 19 | 8 | 1 | 1.5 | 3 | 1 |
| Multivisceral | 20 | 0 | 0 | 18.5 | 34 | 15 |

**Table 8. Median number of tests done per subject, stratified by year and only including tests done within first year post-transplant.**

| Year | Total including deceased | Total excluding deceased | EBV- by PCR | EBV+ by PCR | Liver transplant only | Non-liver transplants |
|---|---|---|---|---|---|---|
| 2007 | 6 | 7.5 | 2 | 9 | 16 | 2 |
| 2008 | 6 | 6 | 0 | 6 | 10.5 | 4 |
| 2009 | 5 | 5 | 4 | 5 | 9 | 2 |
| 2010 | 7 | 8 | 6 | 8 | 10.5 | 2.5 |
| 2011 | 8 | 8 | 9 | 8 | 12.5 | 4.5 |
| 2012 | 7 | 7 | 6 | 7.5 | 11 | 4 |
| 2013 | 11 | 11 | 10 | 13 | 13 | 10 |
| 2014 | 8.5 | 9 | 7 | 11 | 11 | 9 |
| 2015 | 8 | 8 | 7.5 | 8.5 | 9.5 | 7 |
| 2016 | 7 | 7 | 6.5 | 8 | 6.5 | 10 |
| 2017 | 6 | 6.5 | 5 | 9 | 6 | 11 |
| 2018 | 8 | 8 | 7 | 9.5 | 6 | 12 |

and less kidney transplant recipients which is a limitation of this study and this may have affected the hazards ratio. Lung transplants did not have a high incidence of DNAemia as indicated by Table 3, but did have high hazard ratios for PTLD. This again may be due to a small sample size of lung transplant recipients within our population. In addition, many lung transplant recipients from our center are referred back to more local home institutions and EBV PCRs may be obtained externally. Those who stay at our center often have many complications such as rejection, and of course, the development of PTLD. Lastly, type of induction immunosuppression may be important given the trend towards decreased time to PTLD with the use of alemtuzumab, ATG, or high dose steroids. These findings may help with stratifying patients who may be at higher risk for PTLD.

It is not immediately quite clear why risk factors associated with PTLD (Table 2) were different than risk factors associated with increased hazards of PTLD (Table 6). For the analysis of hazard ratios, we did add an extra variable of the maximum level of EBV DNAemia and the Cox regression analysis was adjusted for different variables, and this may have changed the analysis. There may also be statistical differences when examining development of PTLD vs time to PTLD.

Risk factors associated with increased odds of EBV DNAemia were slightly different than risk factors associated with increased hazards of PTLD. This may reflect the fact that while EBV DNAemia is a risk factor for PTLD, how clinical providers manage the EBV DNAemia may influence the development of PTLD. Also, for example, liver transplant recipients were more likely to have EBV DNAemia, but they are less likely to develop PTLD so other factors such as type of organ transplant and immunosuppression may be more important in the development of PTLD.

All pediatric SOT studies for PTLD and EBV are limited by small sample sizes and this single center study is no exception. The overall incidence of PTLD was relatively low. This may have be due to the quick responsiveness of clinicians to high EBV levels. One major limitation is that we were unable to adjust for time-varying exposures to immunosuppressive medications. Higher periods of immunosuppression could have led to increased incidence of PTLD, EBV DNAemia, and higher maximum EBV levels. On the other hand, increased EBV levels may have led clinical providers to decrease immunosuppression, which could have prevented PTLD. However, there is no clear way to delineate all the changes to maintenance immunosuppression as changes occur often and it is very difficult to quantitate such a time-varying risk factor in our analysis.

We did have access to a wealth of data due to frequent EBV monitoring and were able to examine multiple risk factors for EBV DNAemia and the maximum height of EBV DNAemia. However, this study was not adequately powered to examine the numerous interactions between risk factors that may occur in predicting PTLD. Another issue may be the introduction of testing by indication bias, where transplant recipients who had positive and higher levels of EBV were tested more frequently than others. We did note that for the most part, while those who did test positive for EBV by PCR were more likely to receive a higher number of total tests, those who tested negative for EBV by PCR also received a fair number of tests, which denotes that teams were adhering to their protocols (S1 Data).

Other limitations include identifying PTLD and lymphoma cases by diagnosis code and while we did verify each diagnosis, we may have undercounted the number of cases as we may have missed some cases if they were not coded a certain way. Our center uses whole blood sampling for EBV which is much more sensitive but not as specific. While other centers may have moved towards plasma sampling, the more sensitive measure may have led to less cases of PTLD. We did not establish specific cutoffs for the risk of PTLD, but our findings have been shown in a way that is much more useful to individual centers as our final analysis looks at the risk of PTLD associated with each change in log of the quantitative level. We only included EBV quantitative PCRs done at the central laboratory at our center, and we may have missed values that were obtained externally. The decision was made to not include external values in this study as that may have introduced interlaboratory variability to this study. In addition, EBV results from these external laboratories would have been very hard to find in our electronic records, as our center utilized paper charting prior to 2014 and they may not have made their way into each scanned chart from that era.

Despite the study limitations, we did gain some valuable insights into the nature of EBV DNAemia and PTLD in our study population. Further prospective or multicenter studies are needed to link significant clinical factors and EBV DNAemia with PTLD and other clinical outcomes.

## Conclusion

Type of transplant and age were associated with PTLD in pediatric SOT recipients. Age and EBV donor and recipient serology remain key risk factors in the development of EBV DNAemia and the maximum height of EBV DNAemia. Maximum level of EBV DNAemia, and year of transplant were associated with time to PTLD in our population, and there was a trend towards type of induction and time to PTLD. While this study has shed some light on risk factors for EBV DNAemia in pediatric SOT recipients, further studies are needed in order to fully characterize the relationship between EBV levels and PTLD.

## Supporting information

**S1 Data.**
(DOCX)

**S2 Data.**
(CSV)

**S3 Data.**
(CSV)

## Author Contributions

**Conceptualization:** Yeh-Chung Chang, Rebecca R. Young, Alisha M. Mavis, Eileen T. Chambers, Sonya Kirmani, Matthew S. Kelly, Ibukunoluwa C. Kalu, Michael J. Smith.

**Data curation:** Yeh-Chung Chang, Rebecca R. Young.

**Formal analysis:** Yeh-Chung Chang, Rebecca R. Young.

**Methodology:** Yeh-Chung Chang, Rebecca R. Young, Matthew S. Kelly, Ibukunoluwa C. Kalu, Michael J. Smith, Debra J. Lugo.

**Writing – original draft:** Yeh-Chung Chang.

**Writing – review & editing:** Yeh-Chung Chang, Rebecca R. Young, Alisha M. Mavis, Eileen T. Chambers, Sonya Kirmani, Matthew S. Kelly, Ibukunoluwa C. Kalu, Michael J. Smith, Debra J. Lugo.

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
