## [Decision Letter · Decision Letter 0]

11 Jul 2022

PONE-D-22-15480Epstein-Barr Virus DNAemia and Post-Transplant Lymphoproliferative Disorder in Pediatric Solid Organ Transplant RecipientsPLOS ONE

Dear Dr. Chang,

Thank you for submitting your manuscript to PLOS ONE. After careful consideration, we feel that it has merit but does not fully meet PLOS ONE’s publication criteria as it currently stands. Therefore, we invite you to submit a revised version of the manuscript that addresses the points raised during the review process.

ACADEMIC EDITOR: Please revise the manuscript according to Reviewers' comments.

We look forward to receiving your revised manuscript.

Kind regards,

Justyna Gołębiewska

Academic Editor

PLOS ONE

Journal Requirements:

3. PLOS requires an ORCID iD for the corresponding author in Editorial Manager on papers submitted after December 6th, 2016. Please ensure that you have an ORCID iD and that it is validated in Editorial Manager. To do this, go to ‘Update my Information’ (in the upper left-hand corner of the main menu), and click on the Fetch/Validate link next to the ORCID field. This will take you to the ORCID site and allow you to create a new iD or authenticate a pre-existing iD in Editorial Manager. Please see the following video for instructions on linking an ORCID iD to your Editorial Manager account: https://www.youtube.com/watch?v=_xcclfuvtxQ.

Reviewers' comments:

Reviewer's Responses to Questions

**Comments to the Author**

1. Is the manuscript technically sound, and do the data support the conclusions?

Reviewer #1: Partly

Reviewer #2: Yes

2. Has the statistical analysis been performed appropriately and rigorously? 

Reviewer #1: No

Reviewer #2: Yes

3. Have the authors made all data underlying the findings in their manuscript fully available?

Reviewer #1: Yes

Reviewer #2: Yes

4. Is the manuscript presented in an intelligible fashion and written in standard English?

Reviewer #1: Yes

Reviewer #2: Yes

5. Review Comments to the Author

Reviewer #1: Overall, I appreciate the investigators’ efforts to quantify the relative risk for PTLD with increasing EBV DNA loads. I have several comments to improve the manuscript.

1. Introduction, line 52 of pdf: does “type of transplant” mean “type of organ transplant”, as stated in line 56? If yes, please make both sections consistent.

2. Methods, lines 93-94: Billing diagnosis codes are notorious for under-capture of PTLD cases. In the older ICD-9 coding that was in use for a large part of this study’s time period, a diagnosis code for PTLD not even initially present. Did the authors make an additional initial search of their center’s pathology or cancer databases to capture more PTLD cases, using the search term PTLD or its full form, or post-transplant lymphoma? What about from the Problem list of your electronic health record?

3. Methods, line 96: your EBV DNA assay is from whole blood, while most centers have moved to plasma assays. The results from the two sources do not correlate well with each other, with whole blood running higher levels, higher sensitivity but much lower specificity than plasma (see multiple reviews by Preiksaitis J et al). I suggest to add in your Discussion limitations paragraph about the lower generalizability of your results since they are based on whole blood assays only.

4. Methods, lines 100-102: was your EBV PCR DNA assay performed locally or at an outside lab through the 11 year study period? Were there no changes in the assay or the lab used over the 11 years? State in the manuscript what was the conversion factor used from copies/mL to IU/mL.

5. Methods, lines 105-106: why were these specific covariates only used in the logistic regression? Compare with a later paragraph, where you do mention how covariates were selected, but not here. Also, since EBV DNA replication can occur at varying points in time and your patients would have varying periods of follow up, how could you use logistic regression here, versus a Cox proportional hazards model?

6. Results, lines 128-129: your population is skewed heavily towards liver transplants, with very few kidney transplants, unlike the proportions at most pediatric transplant centers. This should also be added in your Discussion limitations paragraph as a factor that lowers generalizability of your results.

7. Table 1, row heading EBV status post-transplant: does this refer to whether recipients turned PCR DNA positive or not? If yes, please make the heading more specific. Similarly, in the text on line 136 “All children who turned PTLD were EBV positive” – is this referring to recipient EBV seropositivity at time of transplant, or recipients who turned EBV PCR DNA positive post-transplant?

8. Table 2: all prior studies have shown that donor recipient EBV seromismatch (D+/R-) associates to an elevated risk for PTLD, and often this risk factor has the highest magnitude of all risk factors (see PMID 29178667). The incongruence of this result is highlighted further in Tables 3 and 4, where D+/R- is the biggest risk factor for EBV DNAemia. You need to state in your Discussion that your result is in discord with prior literature.

9. Results line 155: I think you inverted in the text sentence the ORs, since D+/R- is mentioned first and Odds Ratio 3.90 is mentioned first. (Should be D+/R- with OR 4.80 and D+/R+ with 3.90).

10. Why are the results for risk factors for PTLD (Table 2) different for risk factors for TIME to PTLD (Table 6)?

11. The authors have available to them granular single center data – you should also therefore be able to look at maintenance immunosuppression changes in response to EBV DNA positivity or acute rejection episodes – these would affect your results.

Reviewer #2: Major comments:

1. The authors have described the general algorithm for EBV screening in the cohort, but a more detailed description of EBV PCR tests actually performed during follow up would be helpful, i.e how many tests were performed in total and stratified by transplant type, what was the median number of samples performed during the first vs later years, and were there children who did not have any screening performed during follow up? This is due to the risk of introducing a testing by indication bias. Thus, the difference in the association between EBV DNAemia and risk of PTLD in the different transplant types could be due to different sampling regimens. Recipients with EBV DNAemia may be tested more intensively compared to patients who do not have EBV DNAemia and thus a potential positive test may be missed in those patient and you may then also miss a potential increase in viral load in these patients. Further, EBV serology may be an indicator for the screening plan and thus again risk underestimating positive and high EBV levels.

2. The potential difference in immunosuppression treatment regimens during follow-up and between departments could have an impact on PTLD incidence and thus introduce a bias. Thus, the potential reduction of IS dosage whenever a positive EBV is detected could prevent PTLD occurrence and thus underestimate this diagnosis in some groups. This could be discussed in the limitation section.

3. it is interesting that it seems that you find a lower risk of EBV DNAemia in lung and heart tx sompared to fx kidney. This is in contrast to must studies where lung and heart are more heavily immunosuppressed and thus have a higher risk of developing EBV DNAemia. This could be discussed in the discussion section.

Furter, it is interesting that although Lung tx have very low risk of EBV DNAemia, they have the highest risk of PTLD. How do you explain this?

4. your write that “It is also important to note that an earlier date of 212 transplant was associated with a higher hazards ratio for PTLD. This may have been due to better EBV 213 monitoring over time and the development of protocols which have standardized practice at our center.”

In this case you would also expect an increase in the patients who were tested positive for EBV DNA. Did you see this?

Minor comments:

Table 1: EBV status post transplant: Revise to EBV DNAemia detected posttransplant.

Line 136: all PTLD were EBV positive. Was EBV detected prior to or at time of diagnosis? Some information about the time relation between the detected EBV and PTLD diagnosis would be helpful.

Line 138: Confidence intervals to incidence of PTLD would be helpful. Median time to PTLD?

Table 6: there seems to be a typo in the confidence interval for lung tx where the upper limit is 1957.

6. PLOS authors have the option to publish the peer review history of their article (what does this mean?). If published, this will include your full peer review and any attached files.

Reviewer #1: No

Reviewer #2: No

---

## [Author Response · Author response to Decision Letter 0]

9 Aug 2022

We have also responded to the following points made by the reviewers.

Reviewer #1:

1. In line 52, the term “organ transplant” has been changed to “type of organ transplant” to correspond to similar language in line 56. We have also changed the headings in all of our tables from “Type of Transplant” to “Type of Organ Transplant” in order to stay consistent.

2. Methods, line 93-94: We did identify diagnosis codes for lymphoma in addition to PTLD and thus, this has been added to the manuscript. As for the comment that there was not an ICD diagnosis code for PTLD, there was one from 2008 forward, and thus, we went 1 year previous to this and started our study in 2007. We did not look at a cancer registry as we did not have access to one and we did not look at problem lists as they are notoriously hard to pull from the charts and also come with their shortcomings. We do realize that there are limitations of identifying PTLD and lymphoma by diagnosis code, and have added this to the discussion section. In the end, we wanted to make sure that the cases of PTLD were indeed true cases of PTLD, proven by biopsy. 

3. Methods, line 96: We did use whole blood quantitative EBV PCRs, which as one reviewer mentions is more sensitive but less specific. Indeed, there is much intervariability of quantitative PCRs, and we have included the reference mentioned. In the end, our data is presented not as a specific cutoff but as an association between serial levels using the same laboratory and this may be more universally applicable. This has all been added to the Discussion section as a limitation of this study.

4. Methods, line 100-102: We used only quantitative EBV PCRs from our electronic health system which only pulled in values done at our central laboratory. In checking with the laboratory, they used the same machine over the time period of the study. We also confirmed with them the exact conversion factor, 1 copy/µL = 113.6 IU/mL which has not changed over the study period. All this was added into the manuscript under Methods.

5. We have taken out language regarding logistic regression for PTLD in order to clarify the methods, especially as we do not show these results. Logistic regression was used for the odds of EBV DNAemia, and this is detailed further. In response to how a logistic regression was OK for EBV DNAemia despite the fact that patients develop EBV DNAemia over time, we did notice that a pattern that those transplants who were EBV donor positive/recipient positive (D+/R+) and donor positive/recipient negative (D+/R-) were more likely to develop EBV DNAemia. There were also increased odds of EBV DNAemia for donor negative/recipient positive (D-/R+) subjects although this was not significant. In all of these patients, we noticed that the EBV DNAemia occurred pretty quickly after transplant, and thus, the time to EBV DNAemia was not as variable. As shown in Table 4, not many variables were significantly associated with odds of EBV DNAemia, and the ones that were significant were not time dependent. 

6. Results, line 128-129: We take into account the excellent point made on our patient population. We have added the observation that our sample population is heavy in liver transplant recipients, and has less kidney transplant recipients than the typical pediatric center in the Discussion section.

7. EBV positive clarification: We added PCR positive to clarify our definition of EBV positive in Table 1. As further clarification, we also added EBV positive by quantitative PCR to the description “All patients who developed PTLD were EBV positive” on line 136.

8. The reviewer states that in the literature, often D+/R- is one of the highest risk factors for PTLD, and this was not the case in our study. We have added it to the limitations section of this study. We are not sure why our data does not match this, it could be a sample size issue, something intrinsic to our population, or the fact that providers much more carefully followed patients if they were D+/R-.

9. Results line 155: The odds ratios for EBV D+/R- and D+/R+ were indeed switched and we have edited this to correctly correspond with the proper EBV serostatus category. 

10. We have added more in the Discussion section as to why there are differences between risk factors for PTLD (Table 2), and risk factors for time to PTLD (Table 6). There is an additional variable in the analysis in Table 6, the log¬10 transformed maximum level of EBV. Also, in the analysis in Table 6, each variable is adjusted against the others. Lastly, with the analysis of time to PTLD, we do realize that we cannot study all the variables that vary over time such as immunosuppression and this may confound the results as well. I think it is interesting to see that Table 2 presents the systematic overview of associations and is a good frame of reference, and Table 6 presents variables that may be more important for individualistic decision-making. 

11. While we do have granular EBV quantitative levels, there are too many changes in the patients’ immunosuppressive maintenance regimens over time and we could not study changes in all the patients. While we can pull out T cell depleting agents such as anti-thymocyte globulin (ATG) and alemtuzumab (Campath), it is very hard to categorize all the small changes to maintenance immunosuppression. In our center, changes to medications could reflect a change in levels or an actual adjustment in immunosuppression level. This would require a massive chart review as clinical notes would be the gold standard here, in order to generate a detailed summary of immunosuppression changes, which would be extremely time intensive. Patients do obtain troughs but these values are also variable and do not always reflect true goals. We would have to do a chart review looking at clinical notes and as patient immunosuppressive levels change over time, it would be hard to categorize changes. We do state this as a major limitation of this study. 

Reviewer #2:

1. We understand the issue of testing by indication bias, and have added this to the limitations. This is a very important point. We have additional tables in the Supplementary Data section that does confirm this. Whether stratified by year or type of organ transplant, for the most part, those patients who are EBV positive by PCR have a higher median number of tests when compared to those who are EBV negative PCR. Per our protocol and guidelines, those who have higher levels of EBV DNAemia are monitored more closely and yes, there is a lower threshold to decrease immunosuppression in these patients. However, to the point of sudden EBV DNAemia (primary infection) resulting in PTLD, as you can see from Figure 1, the vast majority of patients who developed PTLD had high EBV levels for about 2 months or so before developing PTLD. There weren’t a lot of patients where the development of PTLD was caught by surprise. We do realize that a better summary of how EBV testing was done can help frame our reference, and we have thus included this extra data in the Supplemental Data section. 

2. We have added the specific comment that teams could decrease immunosuppression as a response to high EBV levels and thus reduce risk for PTLD to our Discussion section. There is no easy way to delineate all the changes in maintenance immunosuppression at our center and as previously mentioned, we have stated this as a major limitation.

3. Lung transplants did not have much EBV DNAemia but had high hazards ratio for PTLD. This is likely due to testing nuances at our center. Lung transplants are followed for only a short time period at our center, and most of them receive EBV levels as external labs which we cannot capture easily. We do have a biased sample of this population because those who stay at our center often are those who are sicker, gone through rejection, and of course, they stay or become referred back into our care if they are diagnosed with PTLD. 

4. We did not see more EBV over time. We do have some raw data on this. What we did see was more frequent monitoring at our center in the heart and kidney transplant recipients, which likely led to immunosuppression adjustments over time. Multivisceral/intestinal transplant recipient EBV monitoring was kept constant, and monitoring in liver transplant recipients actually decreased over time, but we did not see as much PTLD in liver transplant recipients. Please see the Supplemental Data section for this.

Minor Comments:

Table 1: In order to clarify the phrase “EBV positive post-transplant”, we added the words “PCR positive” was added

Line 136: To clarify the phrase “all cases of PTLD were EBV positive”, this was edited to refer to PCR positivity prior to PTLD diagnosis.

Line 138: For the comment on confidence intervals, it is very hard to put a confidence interval on time to PTLD, the curve is non-parametric. Also, with the hazard ratios that had smaller sample size, the confidence interval may be extremely wide and harder to interpret.

Table 6: The 1957 on the CI of the hazards ratio for lung transplants is not an error. Due to the very limited sample size of lung transplants (we only had one positive PTLD case amongst 19 cases, and if you looked at the EBV data for lung transplants, it was very sparse as most are referred back to a closer home institution after transplant for EBV monitoring there.

---

## [Decision Letter · Decision Letter 1]

22 Sep 2022

Epstein-Barr Virus DNAemia and Post-Transplant Lymphoproliferative Disorder in Pediatric Solid Organ Transplant Recipients

PONE-D-22-15480R1

Dear Dr. Chang,

We’re pleased to inform you that your manuscript has been judged scientifically suitable for publication and will be formally accepted for publication once it meets all outstanding technical requirements.

Kind regards,

Justyna Gołębiewska

Academic Editor

PLOS ONE

Additional Editor Comments (optional):

Reviewers' comments:

Reviewer's Responses to Questions

**Comments to the Author**

1. If the authors have adequately addressed your comments raised in a previous round of review and you feel that this manuscript is now acceptable for publication, you may indicate that here to bypass the “Comments to the Author” section, enter your conflict of interest statement in the “Confidential to Editor” section, and submit your "Accept" recommendation.

Reviewer #1: All comments have been addressed

Reviewer #2: All comments have been addressed

2. Is the manuscript technically sound, and do the data support the conclusions?

Reviewer #1: (No Response)

Reviewer #2: Yes

3. Has the statistical analysis been performed appropriately and rigorously? 

Reviewer #1: (No Response)

Reviewer #2: I Don't Know

4. Have the authors made all data underlying the findings in their manuscript fully available?

Reviewer #1: (No Response)

Reviewer #2: Yes

5. Is the manuscript presented in an intelligible fashion and written in standard English?

Reviewer #1: (No Response)

Reviewer #2: Yes

6. Review Comments to the Author

Reviewer #1: (No Response)

Reviewer #2: Thank you for the revisions and aswers.

I believe all comments have been satisfyingly adressed and I have not further comments.

7. PLOS authors have the option to publish the peer review history of their article (what does this mean?). If published, this will include your full peer review and any attached files.

Reviewer #1: No

Reviewer #2: No

---

## [Editor Report · Acceptance letter]

7 Oct 2022

PONE-D-22-15480R1 

Epstein-Barr Virus DNAemia and Post-Transplant Lymphoproliferative Disorder in Pediatric Solid Organ Transplant Recipients 

Dear Dr. Chang:

I'm pleased to inform you that your manuscript has been deemed suitable for publication in PLOS ONE. Congratulations! Your manuscript is now with our production department. 

Kind regards, 

on behalf of

Dr. Justyna Gołębiewska 

Academic Editor

PLOS ONE